# Safety Assessment of Endocrine Disruption by Menopausal Health Functional Ingredients

**DOI:** 10.3390/healthcare9101376

**Published:** 2021-10-15

**Authors:** Soyeon Kang, Hagyeong Jo, Mee-Ran Kim

**Affiliations:** 1St. Vincent’s Hospital, Division of Gynecologic Endocrinology, Department of Obstetrics and Gynecology, College of Medicine, The Catholic University of Korea, Seoul 06591, Korea; ksy@catholic.ac.kr; 2Seoul St. Mary’s Hospital, Department of Obstetrics and Gynecology, College of Medicine, The Catholic University of Korea, Seoul 06591, Korea; jhk91101@naver.com; 3Seoul St. Mary’s Hospital, Division of Gynecologic Endocrinology, Department of Obstetrics and Gynecology, College of Medicine, The Catholic University of Korea, Seoul 06591, Korea

**Keywords:** menopause, functional food, gynecology, phytoestrogens, estrogen receptor alpha, estrogen receptor beta, endocrine disruptors, safety

## Abstract

During menopause, women experience various symptoms including hot flashes, mood changes, insomnia, and sweating. Hormone replacement therapy (HRT) has been used as the main treatment for menopausal symptoms; however, other options are required for women with medical contraindications or without preference for HRT. Functional health foods are easily available options for relieving menopausal symptoms. There are growing concerns regarding menopausal functional health foods because the majority of them include phytoestrogens which have the effect of endocrine disruption. Phytoestrogens may cause not only hormonal imbalance or disruption of the normal biological function of the organ systems, but also uterine cancer or breast cancer if absorbed and accumulated in the body for a long period of time, depending on the estrogen receptor binding capacity. Therefore, we aimed to determine the effects and safety of menopausal functional health ingredients and medicines on the human body as endocrine disruptors under review in the literature and the OECD guidelines.

## 1. Introduction

The average life expectancy of human beings is increasing with improvements in standards of living, economic level, scientific advances, and medical technology. The mean age of menopause is reported as 49.9 years [1], and menopausal women live an average of 30 years or more after menopause [2]. In addition, as women’s education and living standards improve, personal and social interests in the treatment of menopausal symptoms and health care after menopause are increasing. Menopausal transition begins when the function of the ovaries begins to weaken, and it ends when ovarian function disappears completely with loss of female fertility [3,4]. During the period of menopausal transition, which usually begins a few years in advance and typically ends one year after the final menstrual period [3,5], women experience physical symptoms including hot flashes, sweating, muscle pain, and genitourinary symptoms, as well as psychological symptoms such as irritability, nervousness, anxiety, insomnia, and depression [3,6,7]. For these reasons, health management is considered very important for middle-aged women experiencing menopause. Hormone replacement therapy (HRT) has been used as the main treatment for menopausal symptoms, although other options are required for whom HRT is not available because of personal preference or medical contraindications such as hormonally dependent cancers [8]. Prescription therapies such as selective serotonin reuptake inhibitors (SSRIs), serotonin-norepinephrine uptake inhibitors (SNRIs), gabapentin, and clonidine have shown some degree of efficacy as nonhormonal treatment options in women with menopausal symptoms [8,9]. Over-the counter supplements and herbal therapies are also available [8,9] and some are classified as functional health foods on the condition that they are certified for functionality and safety by the Korea Food & Drug Administration (KFDA) [10].

Along with the emerging needs of alternative treatments for symptomatic menopausal women, and also with the social trend of pursuing a healthy life, interest in and sales of functional health foods are steadily increasing. These products help women experiencing menopause by alleviating their symptoms and enabling middle-aged women to live happy and constructive lives [11]. The majority of functional health foods for relieving menopausal symptoms contains phytoestrogens, which are found naturally in foods such as isoflavones, prenylfavonoids, coumestnas, and lignans [9]. Because phytoestrogens are structurally similar to anthropogenic endocrine disruptors and act similarly on numerous molecular and cellular targets, there are growing concerns about the effect of endocrine disruption by phytoestrogens [12].

An endocrine disruptor, also called an endocrine disrupting compound (EDC), is defined as “an exogenous substance that inhibits the synthesis, secretion, body transport, binding, excretion or hormonal action itself “ by the U.S. Environmental Protection Agency (EPA) [13]. As defined, endocrine disruptors influence homeostasis, development, and cell proliferation by mimicking or inhibiting the actions of endogenous hormones, and produce developmental toxicity, carcinogenicity, immunotoxicity, and neurotoxicity [14]. Many endocrine disruptors, including chemicals, pesticides, environmental contaminants, and metals, have been reported to be associated with critical reproductive disorders such as miscarriages, endometriosis, polycystic ovarian syndrome, and infertility. There are also reports of endometrial cancer and breast cancer as adverse effects of endocrine disruptors in female reproductive organs [14].

However, the endocrine-disrupting properties of phytoestrogens are not fully elucidated. One of the features of EDCs is to accumulate in the human body without excretion for long periods of time [14]. Considering this feature in connection with the fact that most functional health foods containing phytoestrogens are easily accessible without a prescription from health professionals in Korea, and that middle-aged women who consume phytoestrogens can live an average of 30 years or more after menopause [2], safety of phytoestrogens must be evaluated and validated; When a woman’s body is exposed to these substances alone or together with other substances from the environment, they can accumulate in the body and have harmful effects on their health [15] in the near or distant future by affecting the uterus and ovaries [16,17].

Therefore, in this review, we aimed to investigate the types of endocrine disruptors in menopausal functional health foods, the extent to which people are exposed to them, and what measures can be taken to protect against their harmful effects. In addition to studies showing the efficacy of plant-derived phytoestrogens and extracts of natural substances on human health, further research is required to determine the presence of endocrine disruptors using methods such as the OECD–provided guidelines.

## 2. Diet and Supplementary Foods for Midlife Women

Healthy diet comprising several daily servings of fruits and vegetables, whole-grain fibers, fish twice/week, and low total fat (such as olive oil) is recommended for midlife and menopausal women as a best practice based on the clinical experience of the members of the International Menopause Society (IMS) [18]. Similarly, the Mediterranean diet is recommended for maintaining health during menopause as a non-restrictive dietary pattern that may improve vasomotor symptoms, mood and depressive symptoms, and cardiovascular risk factors such as blood pressure, cholesterol, and blood glucose level as short-term effects, and may maintain or improve bone mineral density, in according to an EMAS position statement [19]. The Mediterranean diet is a plant-dominant dietary pattern that is low in saturated fat (olive oil and nuts are the major source of fat), and consists of daily consumption of fruit, vegetables, breads, other forms of cereals, beans and seeds, olive oil and nuts, dairy products such as cheese and yogurt, and eggs and fish [19]. Daily limit on salt and alcohol consumption is also recommended.

With regard to complementary foods that contain phytoestrogens such as isoflavone and black cohosh, and St John’s wort, women should be counseled that these components have limited evidence of efficacy and safety [18]. According to the 2015 position statement of the North American Menopause Society (NAMS) for nonhormonal management of menopause-associated vasomotor symptoms, treatment with S-equol derivatives of soy isoflavones and extracts is suggested as a reasonable option with caution for postmenopausal women with mild to moderate hot flashes. Treatment with S-equol supplementation can continue with monitoring for adverse events if a women responds to S-equol supplementation, but other treatment options should be discussed if a women does not respond after 12 weeks [8]. Despite supplemental phytoestrogens being available and promoted to treat menopausal symptoms, there have been few randomized controlled trials of sufficient power and duration to reach conclusions regarding the impact on hot flashes and other menopausal symptoms as well as their long-term effects on the breast, uterus, brain, cardiovascular disease, and thyroid function [9]. Hence, several over-the-counter supplements and herbal therapies, including black cohosh, crinum, dioscorea, dong quai, evening primrose, flaxseed, ginseng, hops, maca, omega-3s, pine bark, pollen extract, puerperia, and Siberian ginseng, are not recommended for use in alleviating vasomotor symptoms in menopausal women [8], although some of these supplements are classified as functional health foods by the KFDA [10]. Moreover, recommendations for or against the use of specific food supplements can be critical for those who have had or are at a high risk of having breast cancer. The Comité de l’Évolution des Pratiques en Oncologie (CEPO) recommended that phytoestrogens, black cohosh, and St. John’s wort should not be used to treat hot flashes for breast cancer survivors due to the risk of drug interactions [20]. CEPO also advised against the use of black cohosh in breast cancer survivors because of the potential risk of recurrence [21]. The American Association of Clinical Endocrinologists (AACE) supported these recommendations for breast cancer survivors in a recent update of its position statement on menopause [9]. 

## 3. Phytoestrogens as Endocrine Disruptors

Substances suspected to contain endocrine disruptors include various industrial chemicals, pesticides and herbicides, organochlorine compounds (dioxins), heavy metals (mercury), hormone-like substances (phytoestrogens) present in plants, synthetic estrogens used as pharmaceuticals, and other food additives. Although endocrine disruptors such as bisphenol A and phthalates are metabolized within 6 hours and excreted, some endocrine disruptors are not easily metabolized and remain in the body for many years, concentrated in the body’s fat and tissues [22]; for example, dichlorodiphenyltrichloroethane and its product dichlorodiphenyldichloroethylene [23]. 

Phytoestrogens are naturally occurring plant compounds that are structurally and/or functionally similar to mammalian estrogens and their active metabolites. Because a plant-based diet has many undeniable ecological and health benefits, the intake of soya phytoestrogens is broadly encouraged and regarded as healthy; however, there is a growing concern regarding side effects from the intake of these compounds, especially regarding issues related to the functions of phytoestrogens as EDCs, because the safety regarding their hormone-disrupting activities has not been sufficiently evaluated [24,25].

The mechanisms by which phytoestrogens cause disorders have been studied [25]. Phytoestrogens affected the activity of enzymes related to estrogen synthesis and metabolism, which are crucial processes along with receptor binding and gene expression in hormonal function, in several in vitro studies [25]. Altered activity of these enzymes can lead to endocrine disruption, resulting in fertility-related problems in premenopausal women, and endometrial hyperplasia or breast cancers in menopausal women [26]. Accordingly, avoiding the intake of high doses of phytoestrogens is recommended [27], although there is no consensus on the definition of such “high” doses [28].

Isoflavone, a representative phytoestrogens, has a structure that is very similar to that of estrogen (Figure 1) [24]. 

Isoflavones, which are contained in several legumes such as soybeans, lentil peas and chickpeas, are increasingly consumed as an alternative to postmenopausal hormone therapy. There is controversy regarding the health benefits of long-term use of isoflavones in postmenopausal women. Improvement of endothelial function and possible slowing down of the progression of subclinical atherosclerosis were reported in several studies [29,30], and side effects including abnormal uterine bleeding, endometrial pathology, and dysmenorrhea were shown in young and middle-aged women who consumed soya at more than 40 g per day to alleviate menopausal symptoms. These side effects were resolved when soya intake was discontinued or reduced [31]. Conclusions cannot yet be reached regarding its effectiveness and appropriate dosage, owing to an absolute lack of research results [32]. 

Lignans are found in flaxseed, whole grains, and certain fruits and vegetables [33]. They are known to have the effect of alleviating menopausal symptoms, as well as antioxidative, anticancer, antibacterial, anti-inflammatory, and cardiovascular protective effects [34,35,36]. Similar to isoflavones, they are a type of phenolic phytoestrogen and can act directly on the estrogen receptor due to their similar chemical structure to estrogen [36]. When a phytoestrogen enters the human body, it binds to the estrogen receptors ERα and ERβ and exhibits estrogen-like activity, and studies have shown that a large amount of phytoestrogens can interfere with the endogenous function of estrogen and affect the female reproductive organs [37].

Genistein, an isoflavone, stimulated the growth of estrogen receptor-positive breast cancer cells implanted in an ovariectomy animal model in a concentration-dependent manner, increasing the cancer risk [38,39]. Therefore, additional research on genistein is needed with inference that the intake of genistein may stimulate the growth of breast cancer in postmenopausal women despite low endogenous estrogen levels. There were also other studies on negative effects such as increased risk of uterine or breast cancer due to phytoestrogens [14]. However, these negative effects are less understood than the harmful effects of industrial EDCs, which have been known despite phytoestrogens being easily found in one’s daily diet, such as in soybean-related products, cereals, beer, and sunflower seeds.

As for the side effects of phytoestrogens as EDCs on postmenopausal women, an increased risk of endometrial hyperplasia and possible goitrogenic activity of thyroid gland [36] were reported. 

The next section focuses on herbal medicines and functional health foods that contain phytoestrogens as a functional ingredient approved by the KFDA to alleviate menopausal symptoms.

## 4. Status of Functional Health Ingredients for Women in Menopause

Estrogen, a female hormone secreted by the ovaries, has a huge impact on disease, and its concentrations are higher in women. Estrogen secretion begins to decrease as the ovaries age, beginning in the mid-30-s. Menopausal symptoms caused by decreasing estrogen include hot flashes, palpitations, sweating, fatigue, anxiety, depression, insomnia, and vaginal dryness [40].

Estrogen acts through two receptors: estrogen receptor alpha (ERα) and estrogen receptor beta (ERβ). ERα is mainly expressed in the uterus and pituitary gland, and ERβ is expressed in ovarian granulosa cells [41], brain [42,43], bones [44,45], and the liver [46,47,48]. 

By consuming functional health food with ingredients that the above-mentioned experiments verified [49], middle-aged women can obtain various benefits: their menopausal symptoms can be alleviated, and their general health condition can be improved [50]. Prevention of osteoporosis is also one of the beneficial effects [51]. However, estrogenic activity induced by binding of phytoestrogens to estrogen receptors can cause endocrine disruption, resulting in side effects including uterine disease, breast cancer, or polycystic ovary syndrome in middle-aged women [52,53]. In particular, excessive estrogenic activity with hormonal imbalance can increase the risk for middle-aged women of breast cancer, considering that most of breast cancer patients suffer from estrogen receptor-positive cancer types [54]. 

Therefore, the development of alternative medicines for treatment of menopausal symptoms requires tests of whether the product induces activation of estrogen receptors or endocrine disruption, and these tests can only be conducted with approval from countries in North America and Europe. The estrogen receptor transcriptional activation assay (stably transfected transcription activation (STTA) assay), TG455, is an example of those tests among the OECD guidelines, and was used to identify one of the main mechanisms that can cause endocrine disruption in relation to harmful effects on the human body [55]. 

The sales of functional health food for menopausal women in Korea was about KRW 1,077.95 billion as of 2019, accounting for about 12.2% of the functional health food market, ranking 6th [56]. For the health of menopausal women in Korea, there are two types of materials identified as functional ingredients, (1) red ginseng, and (2) *Sophora japonica* fruit extract; and nine types of individually approved ingredients, including (1) pomegranate extract, (2) pomegranate juice concentrate, (3) *Schisandra chinensis* extract, (4) Pycnogenol-French maritime pine bark extract, (5) Lactobacillus acidophilus YT1, (6) Thistle complex extract, (7) Rhapontic rhubarb root extract, (8) extract complex of soybean and hop, and (9) complex of *Cynanchum wilfordii* Hemsley, *Phlomis umbrosa* Turczaninow, and *Angelica gigas* Nakai extracts *(CPAE)*. Herbal medicines including black cohosh and St. John’s wort, although not registered as functional foods, are also frequently used for improvement of menopausal women’s health. Studies on each functional ingredient, individually approved ingredients, and herbal medicine about estrogen receptor binding activity are summarized in Table 1.

### 4.1. Herbal Medicines 

#### 4.1.1. Black Cohosh 

Black cohosh is an herb native to eastern North America that has a history of traditional use among Native Americans for the treatment of various symptoms in women, including menopausal symptoms. In clinical studies conducted in Asia and Europe, black cohosh extract was shown to be effective in relieving symptoms in women with menopausal symptoms and is efficacious as tibolone, a hormone preparation [69,70]. 

Black cohosh is composed of the same compounds as triterpene (acetin, cimi-cifugioside, acetylacteol, 27-deoxyactein, cimi-genol, and deoxyacetylacteol) [71]. Among these, 27-deoxyactein is a standardized component [72] and its structure is shown in Figure 2 [73].

Cohosh has been shown to exhibit estrogenic activity, and black cohosh extract significantly increased ER levels in MCF-7 cells, which are commonly used breast cancer cells (Figure 3) [57,74]. 

In an experiment designed to examine the effect on estrogen, the uterus weight of female rats increased according to the dose of black cohosh extract, without increase of the serum estradiol (E2) concentration.

Black cohosh exerts estrogenic effects on the breast and endometrium without affecting the levels of estradiol, FSH, or LH [75]. However, the association of black cohosh with breast cancer is still insufficiently studied [76], and effects such as hepatitis and cholelithiasis with presumed hepatotoxicity were reported as side effects of black cohosh although evidence to support a causal relationship was lacking [77,78]. Although the cause of hepatotoxicity has not been fully elucidated, in vitro and in vivo experimental data indicated mitochondrial damage and subsequent apoptosis [79] to be the cause of this liver injury. Another reported side effect of black cohosh is vaginal bleeding [57].

#### 4.1.2. St. John’s Wort

St. John’s wort (SJW) is widely used with therapeutic effects on depression, relieving psychological and psychosomatic symptoms such as anxiety and insomnia of menopause, and having an anti-inflammatory effect [58,80]. SJW is known to contain various antidepressant-related components including hyperforin (HF), flavonoids, xanthones, and bioflavonoids [81]. Though the mechanism of action is not yet clear [82], decreased expressions of serotonin receptors, increased numbers of 5-HT receptors, and inhibited synaptosomal serotonin uptake by SJW were found during in vitro studies [83].

Estrogenic and cell-proliferative activities of SJW extract and its major bioactive compound HF were reported in an experimental study; SJW and HF increased luciferase activity in MCF-7 cells transfected with ERα or ERβ (Figure 4) [58]. The binding of SJW and HF to ERα, and the binding of SJW to ERβ, are shown in Figure 4a-d. Although HF treatment induced lower breast cancer cell proliferation than 17β-E2 treatment in vitro [58], data on the long-term effect of SJW on the risk of breast cancer, which is a health-related major concern of middle-aged women, are still insufficient. 

Common side effects of SJW are gastrointestinal symptoms, dizziness, confusion, tiredness, and sedation. Serotonin syndrome can also occur due to SJW, with typical symptoms of mental status changes, and autonomic changes, neuromuscular changes, in addition to anxiety, nausea, headache, and diarrhea. Hair loss is reported in rare cases [83].

In addition, a telephone survey of 43 subjects demonstrated that 20 experienced adverse events such as serotonin syndrome, negative food–drug interactions, withdrawal symptoms, and relapse of depression [84]. These side effects indicate the need to be aware of potential risks when patients take SJW.

### 4.2. Functional Ingredients

#### 4.2.1. Red Ginseng

Red ginseng not only improves immunity, but also has various functions, such as fatigue improvement, memory improvement, and relief of menopausal symptoms. In addition, red ginseng extract contains several functional components, and it has been shown that the saponin component in red ginseng restores femur weight, which was reduced due to osteoporosis, in elderly rats and ovariectomized rats via animal in vivo tests [85,86,87]. Moreover, it has been reported that red ginseng extract improves the Kupperman index and overall health conditions [88,89], including emotional anxiety in menopausal women [89]. 

Red ginseng acts as a weak phytoestrogen by binding and activating ER [90]. Among the components of red ginseng, components such as ginsenoside-Rb1, -Rg1, and -Rh1 have estrogenic activity, and ginsenoside Rb1 and Rg1 are known to be detected in large amounts in red ginseng. Ginsenoside-Rb1 bound to ER in a study using MCF-7 cells (Figure 5) [59], and it was confirmed that ginsenoside-Rh1 also bound to ER in the same cell using a luciferase assay [90]. 

Ginsenoside-Rg1 not only binds to ER in MCF-7 cells but is also associated with ERα nuclear translocation and ERα phosphorylation [91,92]. The structure of each component is illustrated in Figure 6 [93].

As adverse events, two cases of vaginal bleeding were reported in postmenopausal women in a study of red ginseng [94]. However, there is inconsistency among studies regarding the effect of red ginseng on hormonal levels or estrogen receptors [10].

#### 4.2.2. Pomegranate Extract and Pomegranate Juice Concentrate

Experiments have established that pomegranate is effective in relieving menopausal symptoms owing to its estrogen content [95]. Both pomegranate extracts and juice concentrates significantly improved menopausal symptoms calculated using the Kupperman index compared to placebo in clinical trials performed in 2008. Menopause symptoms that were alleviated included hot flashes, paresthesia, insomnia, and irritability [96]. Ellagic acid, an indicator component of pomegranate extract and pomegranate juice concentrate, also bound to ERα in HeLa cells [97]. According to the study results, ellagic acid was significantly bound to ERα at concentrations of 10^−9^ M to 10^−7^ M (Figure 7) [60]. The structure of ellagic acid is shown in Figure 8 [98].

#### 4.2.3. Sophora Japonica Fruit Extract

*Sophora japonica* fruit extract is a functional ingredient that contains 60 % alcohol, and it has been confirmed that it significantly improved the Kupperman index and menopausal symptoms in a human application test conducted for 12 weeks in 87 menopausal women [99].

Studies on the binding of *Sophora japonica* fruit extract to estrogen receptors have also been conducted. Dichloromethane extracts of *Sophora japonica* L. were confirmed to bind to the estrogen receptor through a luciferase assay in MCF-7 cells. In this study, it was found that the dichloromethane fraction of *Sophora japonica* L. extract contained approximately 8003.8 mg/kg of the phytoestrogen genistein, and the degree of binding between such fractions at 1 μg/mL and the estrogen receptor was similar to the concentration of 10 μM genistein as described in Figure 9 [61]. The structure of genistein is shown in Figure 10 [100].

#### 4.2.4. Schisandra Chinensis Extract

*Schisandra chinensis* is known to be beneficial for menopausal women because it contains lignans, which are phytoestrogens. *Schisandra chinensis* extract, manufactured by hot water extraction of Schisandra berry, was approved in 2016. It was confirmed that the Kupperman index and Menopause Rating Scale (MRS) score improved when ingested for 12 weeks by 41 menopausal women [101]. Furthermore, when *Schisandra chinensis* extract was used to treat MCF-7 cells at concentrations of 1, 10, and 100 μg/mL, they bound to both ERα and ERβ, and had stronger binding than 17β estradiol at concentrations of 0.01 μmol/L (Figure 11) [62].

#### 4.2.5. Thistle Complex Extract

Thistle complex is extracted from *Cirsium japonicum* var. maackii (Maxim). It is a complex extract of M. matsums and *Thymus vulgaris* L., and is a functional ingredient most recently approved to improve menopausal women’s health. The efficacy of this ingredient, like that of other individually approved ingredients, has been shown through an ovariectomy animal model and clinical study, with alleviation of menopausal symptoms such as hot flashes and improvement of the Kupperman index [102]. Binding of thistle complex extract to ERα was confirmed by measuring the expression of estrogen receptors in uterine tissues of ovariectomized rats by Western blotting (Figure 12) [63,103]. MCF-7 cells were always bound to ERα, even when the thistle complex extract was added [63].

#### 4.2.6. Lactobacillus Acidophilus YT1

*Lactobacillus acidophilus* YT1 is a functional ingredient that was individually approved in 2019 and is the only probiotic among functional health ingredients for menopausal women. In an ovariectomy animal model, RNA was isolated from the shinbone following treatment with *Lactobacillus acidophilus* YT1, and changes in estrogen receptor expression were measured. As a result, it was confirmed that administration of *Lactobacillus acidophilus* YT1 increased the expression level of ERβ, which had been decreased due to ovarian resection [64]. The group that had been administered estradiol at 1 mg per 1kg of body weight (b.w.) was used as a positive control (PC); in each group, the following were administered orally: the low group was administered *Lactobacillus acidophilus* YT1 1×106 CFU/kg b.w. (OVX + Low), the medium group used 1×107 CFU/kg b.w. (OVX + Medium), and the high group was administered 1×108 CFU/kg b.w. (OVX + High).

In a clinical trial on menopausal women, it was confirmed that the intake of *Lactobacillus acidophilus* YT1 improved the Kupperman index and alleviated menopausal symptoms. The quality of life questionnaire (MENQOL questionnaire) showed improvement in the menopausal women’s quality of life [104]. 

#### 4.2.7. Pycnogenol (French Maritime Pine Bark Extract)

Pycnogenol (French maritime pine bark extract) has various functions that are similar to those of red ginseng, and its functions have been approved by the KFDA for three health claims, including free radical removal, blood circulation improvement, and improvement in menopausal women’s health [105]. When this extract was used to treat MC3T3-E1 cells, an osteoblastic cell line, the expression level of ERβ was increased by approximately 2.6 times compared to the control [65]. In addition, the function of improving menopausal women’s health was confirmed through a human clinical trial targeting menopausal women [106,107]. 

#### 4.2.8. Rhapontic Rhubarb Root Extract

Rhapontic rhubarb root extract is a functional ingredient that was individually approved by the KFDA in 2019. Rhapontic rhubarb root extract contains a component called rhaponticin, which can be found in the rhapontic rhubard root and accounts for approximately 5% of the root components. This ingredient has proven its efficacy in enhancing menopausal women’s quality of life and alleviation of menopausal symptoms via human application tests for 6 months [108]. In an in vitro experiment, it was established that rhapontic rhubarb root extract strongly bound with ERβ. In a test using the ERα/β cell-based reporter assay cell, ERβ was strongly activated, with a 36-fold increase compared to the previously reported activity. Similar to phytoestrogens such as genistein and equol, a nonsteroidal estrogen compound, rhapontic rhubarb root extract and rhapontigenin act as full ERβ agonists (Figure 13) [66].

#### 4.2.9. Extract Complex of Soybean and Hops

The extract complex of soybean and hops contains isoflavone from soybean and 8-prenylnaringenin from hops; these are compounds with a structure similar to that of estrogen and that are known to exhibit estrogenic activity. Isoflavone is a representative phytoestrogen, as previously discussed. 8-Prenylnaringenin has the structure shown in Figure 14 [109], and exhibits estrogenic activity (Figure 15) [67].

This ingredient improved the bone metabolism-related index and bone density in an ovariectomized animal model, and the Kupperman index was significantly improved in a human clinical trial [110].

#### 4.2.10. Complex of Cynanchum wilfordii Hemsley, Phlomis umbrosa Turczaninow, and Angelica gigas Nakai extracts (CPAE)

CPAE is a complex extract composed of *Cynanchum wilfordii* Hemsley, *Phlomis umbrosa* Turczaninow, and *Angelica gigas* Nakai. It does not bind to the estrogen receptor, but exhibits an effect similar to that of estrogen. This estrogenic activity without binding to ER was demonstrated in experiments using both HeLa-9903 cells according to OECD test guideline 455 and MCF-7 cells (Figure 16) [68]. 

Improvement of BMD was shown in several in vivo studies with CPAE, and improvement of the Kupperman index, as well as alleviation of menopausal symptoms, were also confirmed in three clinical trials in Korea and the United States [111,112]. 

### 4.3. Safety Study and Side Effects of Functional Ingredients

Black cohosh extract, which is used as a medicine to treat menopausal symptoms in Korea, contains various bioactive substances. As it has pharmacological efficacy, many studies have been conducted on its safety, and many cases of side effects have been reported. According to the National Institutes of Health (NIH), for example, there have been reports of at least 83 cases of hepatic toxicity owing to black cohosh [113]. Black cohosh is contraindicated during pregnancy due to its potential ability to stimulate uterine contractions [114].

Several studies have focused on the safety issues because there are cases in which natural extracts that are effective in alleviating menopausal symptoms contain phytoestrogens. Considering the circumstances as a whole, caution is required when consuming phytoestrogens since they can act as endocrine disruptors. However, with respect to functional health foods, research and data on safety issues are insufficient, even though they are extracts from natural substances, such as black cohosh extracts. 

There also have been studies conducted on isoflavone, a type of phytoestrogen present in soybean foods [115]. The supplemental intake of isoflavone is effective in reducing hot flashes among women experiencing menopause [116]. In addition, a Korean study has shown that isoflavone intake by menopausal women reduces menopausal symptoms such as hot flashes and fatigue [117]. However, another study demonstrated that women with breast cancer should be mindful about high doses of isoflavone intake for significant periods of time, because this can stimulate proliferation of endometrium and breast cells [118,119]. 

From 2015 to 2019, as seen in Table 2, the number of side effects of functional ingredients increased by 22.5% [120], whereas the number of approved ingredients only increased by 3% and the market size increased by 12.8% [120]. Based on the abovementioned statistics, it is reasonable to state that a lack of safety studies on the functional ingredients contributed to the increase in side effect cases, not the increased sales volume of functional ingredients or increased number of approved ingredients.

## 5. Status of Regulations and Tests for Endocrine-Disrupting Substances by Country

Endocrine disruptors have various physical, chemical, and biological properties, yet there is no internationally recognized comprehensive list of them [121]. Whether a substance is regarded as an endocrine disruptor differs depending on the interpretation of each country or institution, and the scope of the definition of endocrine disruptors is gradually expanding, owing to the inclusion of metabolites, degradation products, and isomers of specific substances [122].

OECD has not only developed test guidelines (TGs) and other tools for testing and evaluating endocrine disruptors including natural substances such as phytoestrogens, but has also proposed 150 standardized test methods for identifying endocrine disruptors through continuous research [123]. 

Potential candidates for substances that may be regarded as endocrine disruptors are identified by various organizations such as the World Wildlife Protection Fund (WWF); the US Environmental Protection Agency (EPA); and Japan’s Ministry of Health, Labor and Welfare, and whether the substance is regarded as an endocrine disruptor is determined by conducting a number of tests.

In the United States, the EPA has been conducting a continuous investigational program to protect human health and the environment, and the Endocrine Disruptor Screening and Testing Advisory Committee (EDSTAC) was formed in 1996 [124]. The EPA provides experimental methods for EDC toxicity evaluation, which were created by the Office of Prevention, Pesticides and Toxic Substances (OPPTS) and Office of Chemical Safety and Pollution Prevention (OCSPP), affiliated with the EPA to provide data according to the Toxic Substances Control Act (TSCA); Federal Insecticide, Fungicide, and Rodenticide Act (FIFRA); and Federal Food, Drug, and Cosmetic Act (FFDCA) [125,126].

The Endocrine Disrupter Screening Program (EDSP), which is implemented by the EPA, selects the most influential and potentially influential substances for estrogen systems (E), androgen systems (A), and thyroid hormones (T), among the hormone systems common to vertebrates, including humans. Experimental guidelines are presented as Tier 1 and Tier 2, as specified in FFDCA section 408 (21 U.S.C. 346a) [125,127].

Registration, Evaluation, Authorization and of Restriction of Chemicals (REACH) regulates registration, evaluation, notification, authorization, and restriction according to the amount and risk of chemical substances in the European Union (EU) system [126]. It was introduced to make registration mandatory for all chemical substances (mixtures and products, including chemical substances) manufactured or imported in the EU over 1 ton per year. These chemical substances must be evaluated or approved based on their quantity and properties. In July 2021, eight substances were added, and 219 substances were listed and managed on the list of Substances of Very High Concern (SVHC) [128].

Japan’s Ministry of Economy, Trade, and Industry passed the Act on the Evaluation of Chemical Substances and Regulation of Their Manufacture in 1973, and it was revised in 1986 with the addition of toxicity tests by the Ministry of Health, Labor, and Welfare. In 2003, the Japanese Ministry of Environment conducted a toxicity test. Since then, the Japanese Ministry of Environment; the Japanese Ministry of Economy, Trade and Industry; the Japanese Ministry of Health, Labor and Welfare; and the Japanese Ministry of Environment have all operated the chemical substance screening system. In 2009, the relevant laws were amended to strengthen the management system. 

In 1998, the Korean government held a meeting on endocrine disruption and formed a “countermeasure council” and “professional research council”, and the Korean term for EDC is “endocrine disrupting compounds”.

The Korean Ministry of Environment benchmarked the registration, evaluation and authorization of chemical substances (REACH) of the European Union (EU). The "Act on Chemical Registration and Evaluation", a Korean version of REACH (K-REACH) was established, and it came into effect in 2015 [129]. Since then, the Korean Ministry of Environment began the process of amending of the Act on Chemical Registration and Evaluation, and this amendment came into effect on 1 January 2019 [130].

As the interest in substances that cause endocrine disruptions is increasing around the world, each country is attempting to regulate such substances by providing a list of substances, and guidelines for their evaluation.

The EPA provides information on estrogen receptor activity based on the ToxCast (TM) “Endocrine Receptor (ER) Model” [125]. Genistein and daidzein, which are well-known phytoestrogens, have ER bioactivity based on the ToxCast (TM) ER model [131]. The EU also lists substances that have been identified as endocrine disruptors under REACH. It also lists substances that could be identified as endocrine disruptors under REACH and phytoestrogens are included [132].

Currently, most countries and organizations are evaluating the safety of endocrine disruptors based on the guidelines published by the OECD, and each country has different evaluation methods. In addition, as these test methods involve non-clinical tests, they can only provide limited information on the effects they might have on the human body. In Korea, however, there is a tendency to think that natural materials are safe, regardless of the possibility that these materials could be endocrine disruptors. In addition, there seems to be a lack of awareness of the safety evaluation being performed on natural materials compared to advanced countries such as the United States and Europe.

Therefore, efforts should be made to provide appropriate safety guidelines for testing and evaluating endocrine disruptors in Korea based on the regulations and safety evaluation methods currently being implemented overseas.

## 6. Conclusions

Endocrine disruptors are receiving worldwide attention because of their risks of causing endocrine disorders including reproductive dysfunction. Disruptions owing to EDCs depend on the chemical structure or its type, and the exact mechanism of most of these substances has not yet been identified.

Individually approved functional ingredients that help improve the health of menopausal women in Korea have proven their efficacy by improving the Kupperman index and menopausal symptoms through human clinical trials for menopausal women [133]. They are sold as functional health foods through sales channels such as online and home shopping. In vitro and in vivo tests were performed, and the efficacy of the functional ingredients for menopause was verified by confirming bone-related indicators, estrogen activity, and ER binding [66,90]. Each extract component prepared with red ginseng, pomegranate, and *Sophora Japonica* bound directly to ER, and other ingredients of functional health foods for menopausal women activated ER in experimental studies [60,61,90]. Currently, CPAE is the only ingredient among functional health ingredients for menopausal symptoms in Korea that helps to improve menopausal symptoms without binding to the ER. Binding to the ER itself does not necessarily mean that they are harmful to the human body, because the dosage of the substances with estrogen-like activity may be small. However, the accumulation of phytoestrogens and other similar endocrine disruptors in the body can cause endocrine disturbances such as reproductive dysfunction or other side effects. Despite studies and tests that have been conducted and regulations that were enhanced for EDCs, including phytoestrogens, the evidence on the safety of phytoestrogens is insufficient due to different regimens, dosages, and durations of follow-up among different studies.

Therefore, middle-aged women should be careful not to consume substances such as phytoestrogens for long periods of time even though they have beneficial effects on relieving menopausal symptoms. It is necessary to verify the safety of such substances via application of international regulations and guidelines to determine whether it is safe to consume them. In the case of menopausal functional health ingredients, we can obtain safety information through review of the degree of estrogen receptor binding by applying OECD guideline 455 [123]. Of course, further studies are needed on the endocrine-disruption effects of functional health foods in menopausal women. Healthcare professionals should be aware of this, and should suggest an appropriate guide for middle-aged women.

## Figures and Tables

**Figure 1 healthcare-09-01376-f001:**
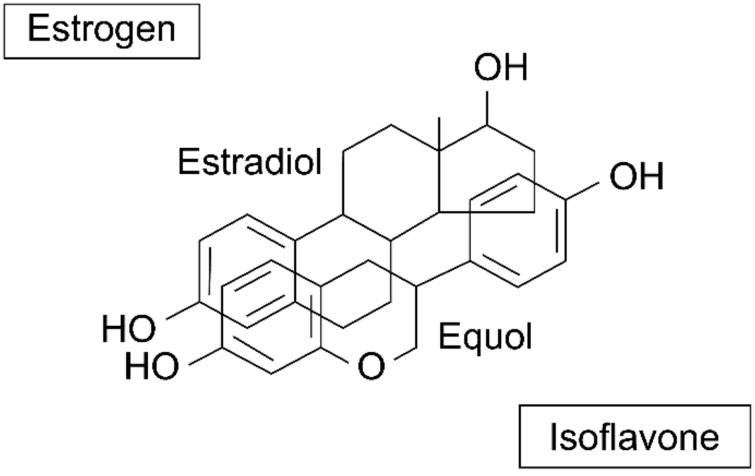
Similarity of isoflavone to estrogen. Reprinted from Setchell, K.D.; Cassidy, A. Dietary Isoflavones: Biological Effects and Relevance to Human Health. *J. Nutr.*
**1999**, *129*, 758–767, with permission. *© 1999, Oxford University Press* [24].

**Figure 2 healthcare-09-01376-f002:**
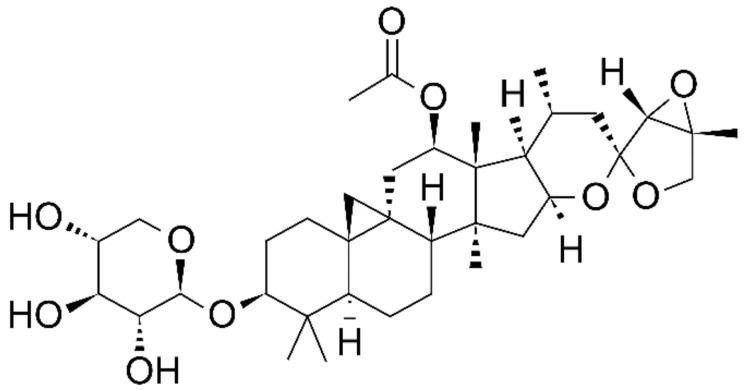
Chemical structure of 27−deoxyactein.

**Figure 3 healthcare-09-01376-f003:**
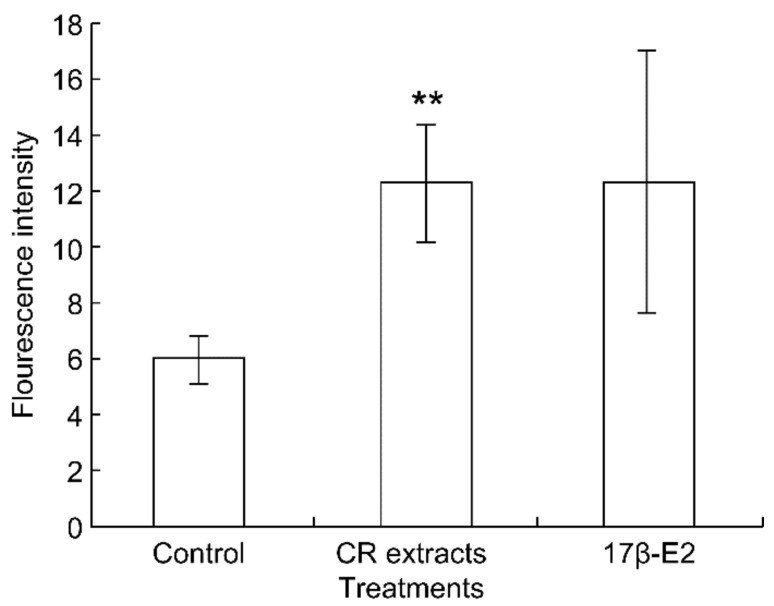
Effects of Black Cohosh extract (CR extracts) on estrogen receptor levels in MCF-7 cells. In comparison with the control group, the results of *t* test are expressed as follows: ** *p* < 0.01. Reprinted from Liu, Z.P.; Yu, B.; Huo, J.S.; Lu, C.Q.; Chen, J.S. Estrogenic Effects of Cimicifuga racemosa (Black Cohosh) in Mice and on Estrogen Receptors in MCF-7 Cells. *J. Med. Food*
**2001**, *4*, 171–178, with permission. *©* 2001, *Mary Ann Liebert, Inc.* [57].

**Figure 4 healthcare-09-01376-f004:**
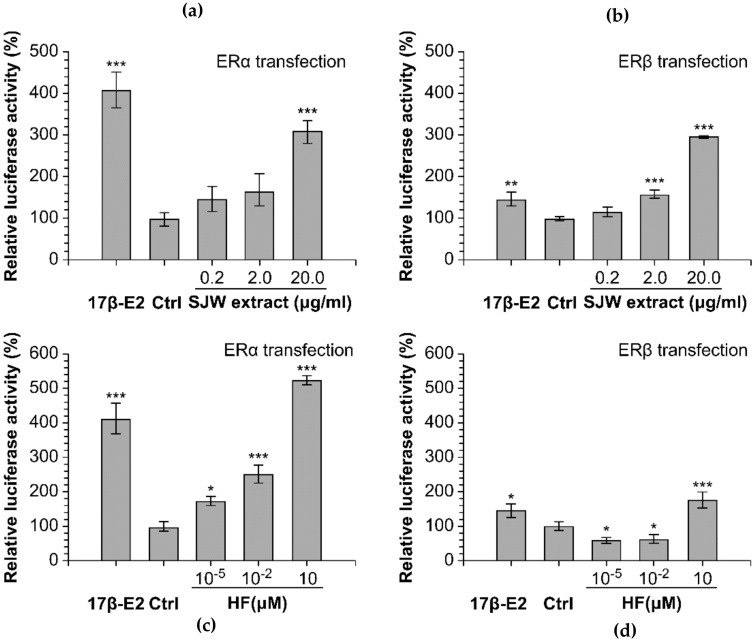
Effect of St. John’s wort (SJW) extract and hyperforin (HF) on estrogen receptor levels in MCF-7 cells. The luciferase activities in MCF-7 cells transfected with ERα (**a**,**c**) or ERβ (**b**,**d**) were measured after treatment with 0.2 to 20 µg/mL SJW extract (**a**,**b**) or 10^−5^ to 10 µM HF (**c**,**d**). A positive control was prepared by treating the cells with 10^−2^ µM 17β-E2. Bars represent the averages of triplicate determinations. * *p* < 0.05 and *** *p* < 0.001. Reprinted from Kwon, J.; Oh K.S.; Cho, S.Y.; Bang, M.A.; Kim, H.S.; Vaidya, B.; Kim, D. Estrogenic Activity of Hyperforin in MCF-7 Human Breast Cancer Cells Transfected with Estrogen Receptor. *Planta Med.*
**2016**, *82*, 1425–1430, with STM permission *© Georg Thieme Verlag KG.* Any further reuse will need explicit permission from publisher [58].

**Figure 5 healthcare-09-01376-f005:**
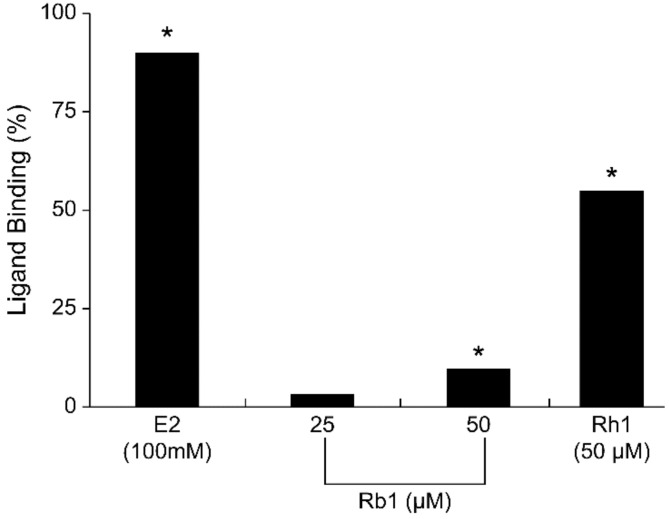
Binding of ginsenoside-Rb1 to ER in MCF−7 cells. * *p* < 0.05. Reprinted from Cho, J.; Park, W.; Lee, S.; Ahn, W.; Lee, Y. Ginsenoside-Rb1 from Panax ginseng C.A. Meyer Activates Estrogen Receptor-alpha and -beta, Independent of Ligand Binding. J. Clin. Endocrinol. Metab. **2004**, *89*, 3510–3515, with permission. © *2004, Oxford University Press* [59].

**Figure 6 healthcare-09-01376-f006:**
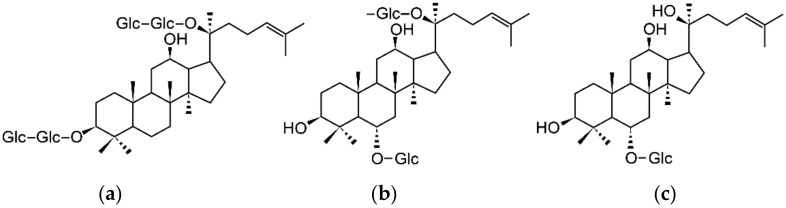
The chemical structures of (**a**) ginsenoside-Rb1, (**b**) ginsenoside-Rg1, and (**c**) ginsenoside-Rh1. Reprinted from Jo J.J.; Shrestha R.; Lee S. Inhibitory Effects of 12 Ginsenosides on the Activities of Seven Cytochromes P450 in Human Liver Microsomes. *Mass Spectrom. Lett.*
**2016**, *7*, 106–110, under the terms of the Creative Commons Attribution License [93].

**Figure 7 healthcare-09-01376-f007:**
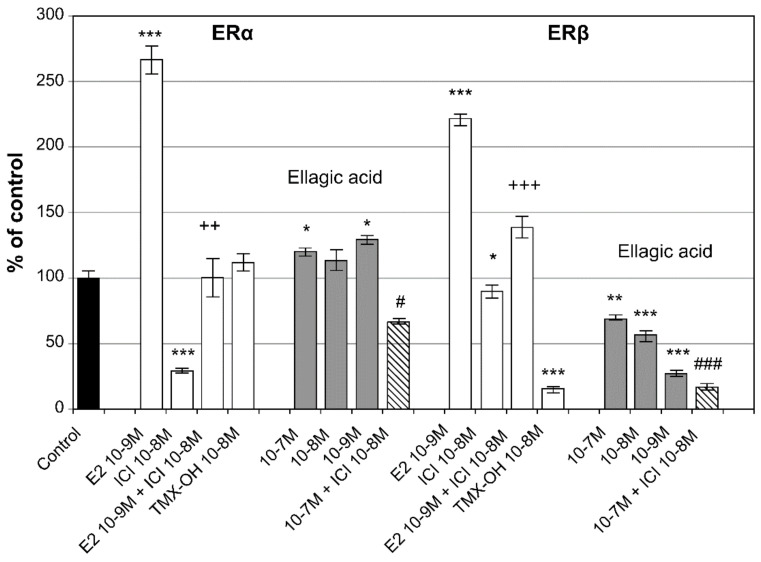
Effect of ellagic acid on estrogen receptor levels in HeLa cells transfected with EREs-containing reporter plasmid with ERα or ERβ, respectively. * Significantly different from control (* *p* < 0.05, ** *p* < 0.01, *** *p* < 0.001). ^+^ Significantly different from E_2_ 10^−9^ M (^++^
*p* < 0.01, ^+++^
*p* < 0.001). ^#^ Significantly different from ICI182780 10^−8^ M (^#^
*p* < 0.05, ^###^
*p* < 0.001). Reprinted from Papoutsi, Z.; Kassi, E.; Tsiapara, A.; Fokialakis, N.; Chrousos, G.P.; Moutsatsou, P. Evaluation of Estrogenic/Antiestrogenic Activity of Ellagic Acid via the Estrogen Receptor Subtypes ERα and ERβ. *J. Agric. Food Chem.*
**2005**, *53*, 7715–7720, with permission. *© 2005, American Chemical Society* [60].

**Figure 8 healthcare-09-01376-f008:**
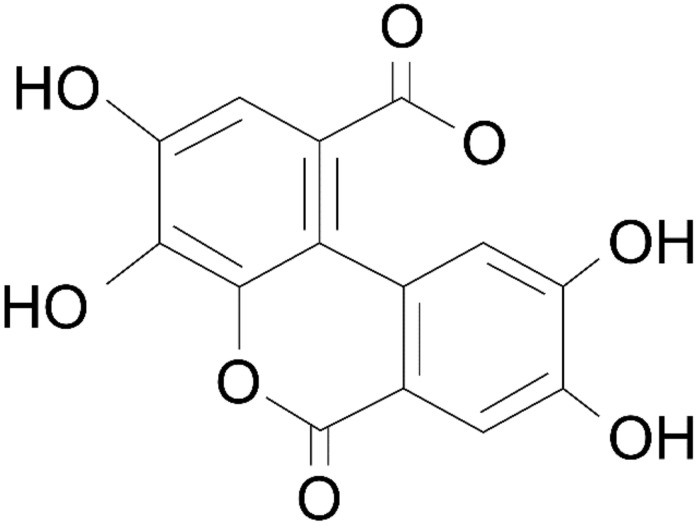
The chemical structure of ellagic acid.

**Figure 9 healthcare-09-01376-f009:**
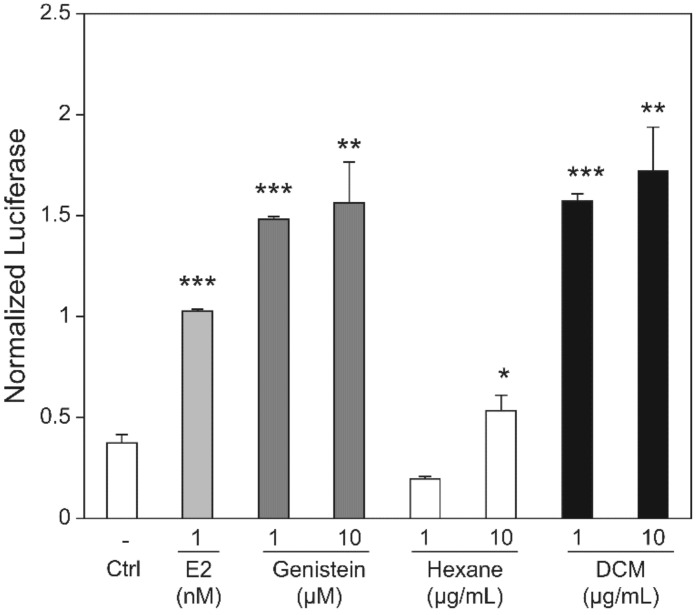
Effects of estradiol, genistein, hexane fraction, and dichloromethane (DCM) fractionated extracts on luciferase activity in MCF-7 cells (* *p* < 0.05; ** *p* < 0.005; *** *p* < 0.0005) Reprinted from Yoon, H.J.; Seo, C.R.; Kim, M.; Kim, Y.J.; Song, N.J.; Jang, W.S.; Kim, B.J.; Lee, J.; Hong, J.W.; Nho, C.W., et al. Dichloromethane extracts of Sophora japonica L. Stimulate Osteoblast Differentiation in Mesenchymal Stem Cells. *Nutr. Res.*
**2013**, *33*, 1053–1062, with permission from *Elsevier* [61].

**Figure 10 healthcare-09-01376-f010:**
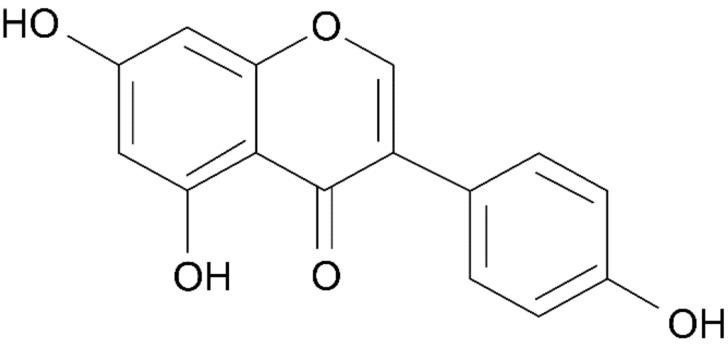
The chemical structure of genistein.

**Figure 11 healthcare-09-01376-f011:**
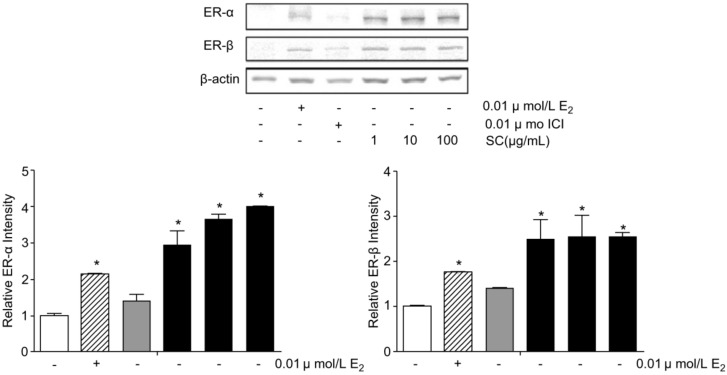
Expressions of ERα and ERβ levels by Schisandra chinensis extract treatment in MCF−7 cells. * *p* < 0.05, compared with normal group. Reprinted from Kim, M.H.; Lee, H.S.; Hong, S.B.; Yang, W.M. Schizandra chinensis Exhibits Phytoestrogenic Effects by Regulating the Activation of Estrogen Receptor-α and -β. *Chin. J. Integr. Med.*
**2017**, 1–5, with permission. *© 2017, Springer Nature* [62].

**Figure 12 healthcare-09-01376-f012:**
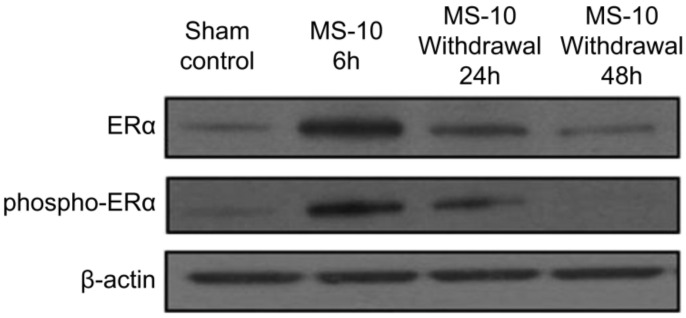
Regulation of ERα expression and phosphorylation by thistle complex extract (MS-10) Reprinted from Noh, Y.H.; Kim, D.H.; Lee, S.A.; Yin, X.F.; Park, J.; Lee, M.Y.; Lee, W.B.; Lee, S.H.; Kim, J.K.; Kim, S.S., et al. The Natural Substance MS-10 Improves and Prevents Menopausal Symptoms, Including Colpoxerosis, in Clinical Research. *J. Med. Food*
**2016**, *19*, 228–237, with permission. *© 2016, Mary Ann Liebert, Inc.* [63].

**Figure 13 healthcare-09-01376-f013:**
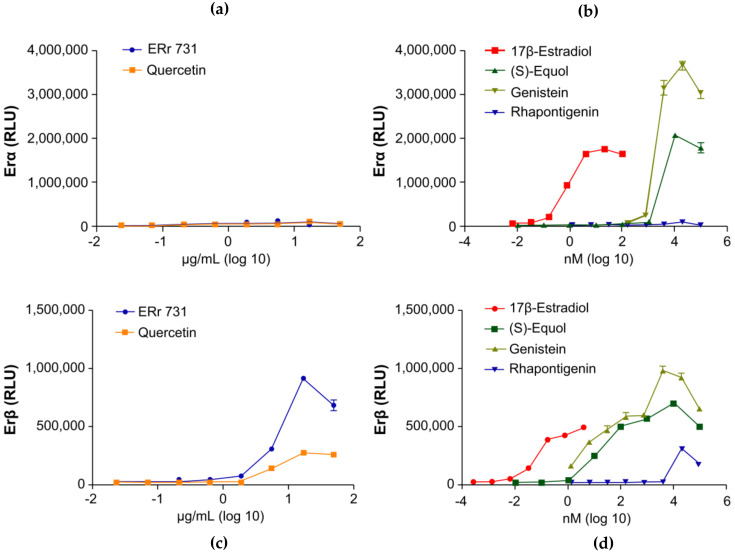
Effect of Rhapontic rhubarb root extract (ERr731) on activity of (**a**,**b**) ERα or (**c**,**d**) ERβ compared to quercetin, estradiol, equol, genistein, and rhapontigenin. Reprinted from Wilson, M.; Konda, V.; Heidt, K.; Rathinasabapathy, T.; Desai, A.; Komarnytsky, S. Rheum rhaponticum Root Extract Improves Vasomotor Menopausal Symptoms and Estrogen-Regulated Targets in Ovariectomized Rat Model. *Int. J. Mol. Sci.*
**2021**, *22*, 1032. *© 2021 by the authors. Licensee MDPI, Basel, Switzerland*. under the terms and conditions of the Creative Commons Attribution license [66].

**Figure 14 healthcare-09-01376-f014:**
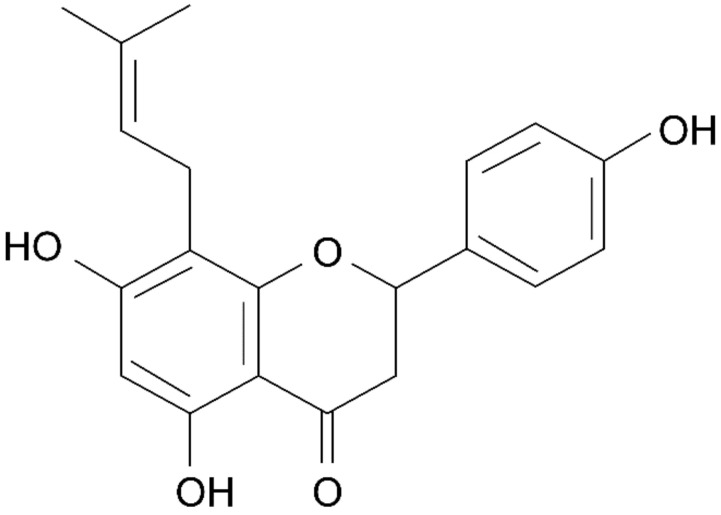
The chemical structure of 8–prenylnaringenin. Reprinted from Jelle, D.K.; Geert, O.; Arne, H., et al. Formation and Accumulation of α-Acids, β-Acids, Desmethylxanthohumol, and Xanthohumol during Flowering of Hops (Humulus lupulus L.), with permission. Copyright © 2003, *American Chemical Society* [109].

**Figure 15 healthcare-09-01376-f015:**
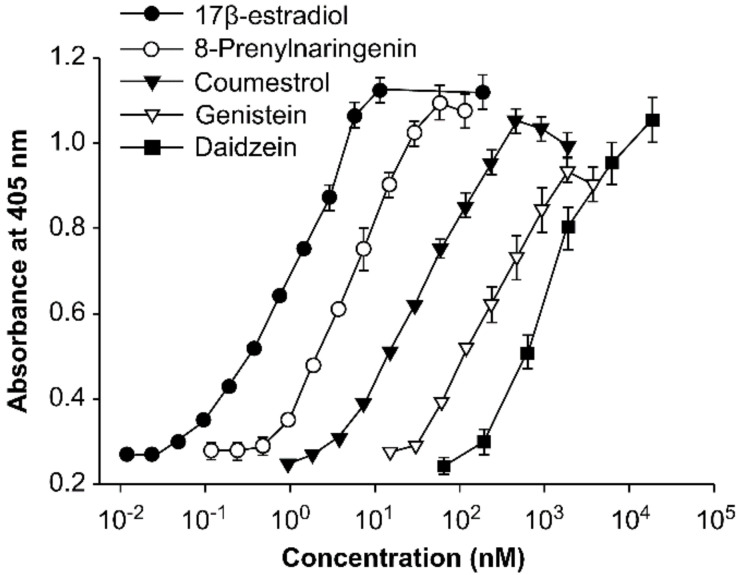
Estrogenic activity of 8–prenylnaringenin and phytoestrogens in human endometrial adenocarcinoma (Ishikawa VarⅠ) cells. Reprinted from Milligan, S.R.; Kalita, J.C.; Heyerick, A.; Rong, H.; De Cooman, L.; De Keukeleire, D. Identification of a Potent Phytoestrogen in Hops (Humulus lupulus L.) and Beer. *J. Clin. Endocrinol. Metab.*
**1999**, *84*, 2249–2252, with permission. *© 1999, Oxford University Press* [67].

**Figure 16 healthcare-09-01376-f016:**
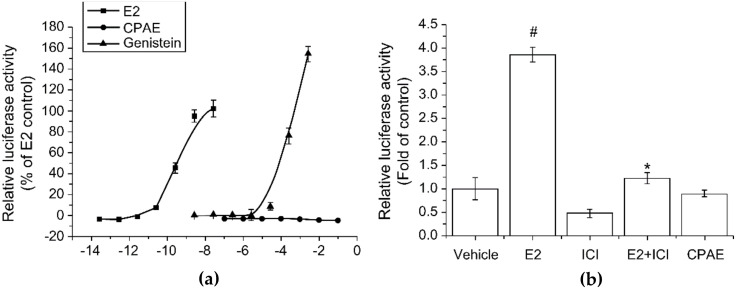
Estrogenic activity of CPAE using luciferase promoter activity in (**a**) HeLa–9903 cells and (**b**) MCF–7 cells, respectively. Bars represent mean ± SD. # *p* < 0.05 vs. control; and * *p* < 0.005 vs. E2. Reprinted from Kim, S.J.; Jin, S.W.; Lee, G.H.; Kim, Y.A.; Jeong, H.G. Evaluation of Estrogenic Activity of Extract from the Herbal Mixture *Cynanchum wilfordii* Hemsley, *Phlomis umbrosa* Turczaninow, and *Angelica gigas* Nakai. *Toxicol. Res.*
**2017**, *33*, 71–77. under the terms of the Creative Commons Attribution Non-Commercial License (http://creativecommons.org/licenses/by-nc/3.0, accessed on 8 August 2021) [68].

**Table 1 healthcare-09-01376-t001:** ER binding activity of the functional health ingredients for women in menopause.

Study	Ingredients	In Vitro/In Vivo Model	ER- α or –β Binding Activity against Control
Liu et al. (2001) [57]	Black cohosh	MCF-7 cell	+
Kwon et al. (2016) [58]	St. John’s wort	MCF-7 cell	+
Cho et al. (2004) [59]	Ginsenoside-Rb1 ^1^	MCF-7 cell	+
Papoutsi et al. (2005) [60]	Ellagic acid ^2^	HeLa cell	+
Yoon et al. (2013) [61]	Dichloromethane extracts of Sophora japonica L.	MCF-7 cell	+
Kim et al. (2017) [62]	Schisandra chinensis extract	MCF-7 cell	+
Noh et al. (2016) [63]	Thistle complex extract	OVX-induced rats	+
Lee et al. (2019) [64]	Lactobacillus acidophilus YT1	OVX-induced rats	+
N et al. (2013) [65]	Pycnogenol	MC3T3-E1 cell	+
Wilson et al. (2021) [66]	Rhapontic rhubarb root extract	Human endometrial cell	+
Milligan et al. (1999) [67]	8−Prenylnaringenin ^3^	Ishikawa Var I cell	+
Kim et al. (2017) [68]	CPAE ^4^	HeLa-9903 cell	-

+, Results of binding activity, *p*-value < 0.05; ^1^ the marker compound of Red ginseng; ^2^ the marker compound of pomegranate; ^3^ the marker compound of extract complex of soybean and hops; ^4^ complex of *Cynanchum wilfordii* Hemsley, *Phlomis umbrosa* Turczaninow, and *Angelica gigas* Nakai extracts.

**Table 2 healthcare-09-01376-t002:** Market size and number of approved cases compared to adverse cases of functional health foods in Korea.

	Year					CAGR ^1^
	2015	2016	2017	2018	2019	
Market size (KRW 100 million)	17,326	20,176	21,297	23,962	28,081	12.83%
Adverse cases, n.	502	696	874	964	1132	22.50%
Cumulative approved cases, n.	560	581	587	601	632	3.00%
Adverse cases, n./Cumulative approved cases, n.	0.90	1.20	1.49	1.60	1.79	

^1^ CAGR, compound annual growth rate; n., number.

## Data Availability

Not applicable.

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
