# Peer review of "Safety Assessment of Endocrine Disruption by Menopausal Health Functional Ingredients"

_healthcare, 2021, doi:10.3390/healthcare9101376_

Round 1

Reviewer 1 Report

The authors of the manuscript entitled “Safety assessment of endocrine disruption by menopausal health functional ingredients” reviewed the types of endocrine disruptors in menopausal functional health foods, the extent of exposure, and measures that can be taken to protect against their harmful effects. The subject of the article is novel, however, some drawbacks make it unfit for publication in this form. The article has the potentials to be accepted provided the authors can address the critical issues listed below.

  1. The aim of the article as stated in the abstract and last paragraph of the introduction is not reflected in the entire manuscript. The authors need to provide commonly recommended food/diet for menopausal women, component endocrine-disrupting chemicals or phytoestrogen, and their harmful effects.
  2. Subheading 2 (Phytoestrogen as an Endocrine Disruptor). Although the heading addresses phytoestrogen, the larger percentage of this section deals with environmental chemicals. The authors should separate environmental chemicals such as dioxins and DDTS from phytoestrogen (See PMID: 33920428, PMID: 34004580 for the appropriate classification of environmental chemicals)
  3. Lines 89 and 90. “Unlike biological hormones, endocrine disruptors are not easily metabolized and remain in the body for many years concentrated in the body's fat and tissues”. This statement is not correct. EDCs such as BPA and Phthalates are metabolized within 6 hours and excreted. Authors should read the literature and correct this statement.
  4. Line 196 -197, 1077.95 billion?
  5. The article contained several published figures and data. Have authors obtained permission from the original authors and publishers to use these figures as they appeared in the original manuscript? If not, all the figures from other publications should be deleted and summarize their findings with reference
  6. Figure 7 should be properly labelled.

Author Response

Dear Reviewer:

We would like to express our sincere thanks to you for your thorough review of our manuscript and for the opportunity to submit a revised and improved version. Your comments were very helpful for revising and improving our manuscript.

We have carefully addressed all your comments in our revised manuscript. The main corrections and point-by-point responses to your comments are provided below, with all corresponding changes highlighted in the manuscript.

We hope that our responses and revisions have adequately addressed your concerns, and that the current version of the manuscript will now meet the high standards required for publication in your esteemed journal. We would be willing to make additional changes should they be required.

Thank you.

1. The aim of the article as stated in the abstract and last paragraph of the introduction is not reflected in the entire manuscript. The authors need to provide commonly recommended food/diet for menopausal women, component endocrine-disrupting chemicals or phytoestrogen, and their harmful effects.

→ We appreciate your important comment. We added a section of “2. Diet and Supplementary Foods for Midlife Women” after introduction and provided food/diet recommendations for menopausal women, phytoestrogens, and their potential harmful effect.

2. Subheading 2 (Phytoestrogen as an Endocrine Disruptor). Although the heading addresses phytoestrogen, the larger percentage of this section deals with environmental chemicals. The authors should separate environmental chemicals such as dioxins and DDTS from phytoestrogen (See PMID: 33920428, PMID: 34004580 for the appropriate classification of environmental chemicals)

→ Thank you for your comment. We removed the part of environmental chemicals from subheading 2, because these contents can distract the theme of this review.

3. Lines 89 and 90. “Unlike biological hormones, endocrine disruptors are not easily metabolized and remain in the body for many years concentrated in the body's fat and tissues”. This statement is not correct. EDCs such as BPA and Phthalates are metabolized within 6 hours and excreted. Authors should read the literature and correct this statement.

→ Thank you for good point. We corrected the sentence like this;

“Though endocrine disruptors such as bisphenol A and phthalates are metabolized within 6 hours and excreted, some endocrine disruptors are not easily metabolized and remain in the body for many years concentrated in the body's fat and tissues. Dichlorodiphenyltrichloroethane (DDT) and its product dichlorodiphenyldichloroethylene (DDE) are such examples.”

4. Line 196 -197, 1077.95 billion?

→ Thank you for your comment. We corrected this as “1,077.95 billion won.”

5. The article contained several published figures and data. Have authors obtained permission from the original authors and publishers to use these figures as they appeared in the original manuscript? If not, all the figures from other publications should be deleted and summarize their findings with reference

→ Thank you for your valuable comment. We have got permissions from the original publishers and stated about the reuse of the figures, except for those of chemical structures or those from open access article distributed under the terms and conditions of the Creative Commons Attribution license. We removed two figures (figure 13 and figure 14 of initial submission) because we could not get permissions for these figures. We also added a summarized table on the studies of each functional ingredient, individually approved ingredient, and herbal medicine in the last section of subheading “3. Status of functional health ingredients for women in menopause”.

6. Figure 7 should be properly labelled.

→ We corrected the label of Figure 7. Thank you for the comment.

Reviewer 2 Report

Recommend to add the reference numbers in the end of Figure legends (Figure 3, Fig 4, Fig 5, Fig 7, Fig 9, Fig 11, Fig 12, Fig 13, Fig 14, Fig 15, Fig 17, Fig 18) because it could mislead that the authors performed these experiments.

Author Response

Title: Safety assessment of endocrine disruption by menopausal health functional ingredients

Dear Reviewer:

We would like to express our sincere thanks to you for your thorough review of our manuscript and for the opportunity to submit a revised and improved version. Your comment was very helpful for revising and improving our manuscript.

We have carefully addressed your comment in our revised manuscript. The main corrections and point-by-point response to your comment is provided below, with all corresponding changes highlighted in the manuscript.

We hope that our responses and revisions have adequately addressed your concerns, and that the current version of the manuscript will now meet the high standards required for publication in your esteemed journal. We would be willing to make additional changes should they be required.

Thank you.

Comments and Suggestions for Authors

Recommend to add the reference numbers in the end of Figure legends (Figure 3, Fig 4, Fig 5, Fig 7, Fig 9, Fig 11, Fig 12, Fig 13, Fig 14, Fig 15, Fig 17, Fig 18) because it could mislead that the authors performed these experiments.

→ Thank you for your valuable comment. We added the reference numbers in the end of the statement for figure reuse and permission, because we have got permissions from the original publishers except for those of chemical structures or those from open access article distributed under the terms and conditions of the Creative Commons Attribution license. We removed two figures (figure 13 and figure 14 of initial submission) because we could not get permissions for these figures.

Round 2

Reviewer 1 Report

The authors have satisfactorily addressed all issues raised regarding the manuscript.

This manuscript is a resubmission of an earlier submission. The following is a list of the peer review reports and author responses from that submission.